# Avian Metapneumovirus Subgroup C Induces Mitochondrial Antiviral Signaling Protein Degradation through the Ubiquitin-Proteasome Pathway

**DOI:** 10.3390/v13101990

**Published:** 2021-10-04

**Authors:** Lei Hou, Xiaohan Hu, Jinshuo Guo, Rong Quan, Li Wei, Jing Wang, Jiangwei Song, Jue Liu

**Affiliations:** 1Beijing Key Laboratory for Prevention and Control of Infectious Diseases in Livestock and Poultry, Institute of Animal Husbandry and Veterinary Medicine, Beijing Academy of Agriculture and Forestry Sciences, Beijing 100097, China; hlbj09@163.com (L.H.); qrcau@126.com (R.Q.); w_lyx2008@126.com (L.W.); jingjing_0047@sina.com (J.W.); songjiangwei525@126.com (J.S.); 2College of Veterinary Medicine, Yangzhou University, Yangzhou 225009, China; Jane9407@163.com; 3Jiangsu Co-Innovation Center for Prevention and Control of Important Animal Infectious Diseases and Zoonoses, Yangzhou University, Yangzhou 225009, China; 4College of Animal Sciences (College of Bee Science), Fujian Agriculture and Forestry University, Fuzhou 350002, China; hxh15855171300@163.com

**Keywords:** MAVS, aMPV/C, ubiquitination, degradation, E3 ubiquitin ligase

## Abstract

The mitochondrial antiviral signaling (MAVS) protein, a critical adapter, links the upstream recognition of viral RNA to downstream antiviral signal transduction. However, the interaction mechanism between avian metapneumovirus subgroup C (aMPV/C) infection and MAVS remains unclear. Here, we confirmed that aMPV/C infection induced a reduction in MAVS expression in Vero cells in a dose-dependent manner, and active aMPV/C replication was required for MAVS decrease. We also found that the reduction in MAVS occurred at the post-translational level rather than at the transcriptional level. Different inhibitors were used to examine the effect of proteasome or autophagy on the regulation of MAVS. Treatment with a proteasome inhibitor MG132 effectively blocked MAVS degradation. Moreover, we demonstrated that MAVS mainly underwent K48-linked ubiquitination in the presence of MG132 in aMPV/C-infected cells, with amino acids 363, 462, and 501 of MAVS being pivotal sites in the formation of polyubiquitin chains. Finally, E3 ubiquitin ligases for MAVS degradation were screened and identified and RNF5 targeting MAVS at Lysine 363 and 462 was shown to involve in MAVS degradation in aMPV/C-infected Vero cells. Overall, these results reveal the molecular mechanism underlying aMPV/C infection-induced MAVS degradation by the ubiquitin-proteasome pathway.

## 1. Introduction

Avian metapneumovirus (aMPV), previously known as avian pneumovirus (APV), is considered an important pathogen to both turkeys and chickens as it causes respiratory tract disease responsible for economic losses to the poultry industry globally [1,2]. aMPV, a member of the family *Paramyxoviridae*, has a single-stranded, nonsegmented, negative-sense RNA genome [3]. Based on the diverse genetic and antigenic properties among the aMPV strains, four subgroups (A, B, C, and D) were isolated from different countries [4]. aMPV subgroup C (aMPV/C) infection in turkeys was first reported in the United States in 1999 and subsequently identified in other states of the USA and in France [5,6,7]. This virus has also been reported and isolated in pheasants in South Korea and in meat-type chickens in China [8,9]. Interestingly, aMPV/C has closer genetics and antigenicity to human metapneumovirus (hMPV) than it does to the aMPV subgroups A, B, and D [10,11,12].

MAVS, a crucial host adapter protein localized on the mitochondrial outer membrane, includes an N-terminal caspase recruitment domain (CARD), a middle proline-rich region (PRR), and a C-terminal transmembrane (TM) domain [13], which is essential for mediating antiviral activity and producing proinflammatory cytokines during viral infection [14,15]. Viral infection induces upstream signal molecules to interact with MAVS, which leads to activation of interferon regulatory factors 3 and 7 (IRF3/7) and nuclear transcription factor-κB (NF-κB) and ultimately induces production of antiviral molecules and proinflammatory factors, which inhibits viral proliferation [16,17].

MAVS is an upstream signal molecule in antiviral innate immunity, which is directly or indirectly regulated by some factors at the transcriptional and post-translational levels instead of interferon (IFN) [18]. At the transcriptional level, reactive oxygen species (ROS) play an important role in the regulation of MAVS mRNA [19]. At the post-translational level, many viruses have various strategies to regulate MAVS function to evade host antiviral mechanisms. Among them, direct cleavage of the MAVS protein is an important aspect. For example, small RNA viruses (Coxsackievirus B3, Seneca Valley virus, and human rhinovirus C) encode a 3C cysteine protease, which cleaves MAVS protein at a specific site and reduces downstream signaling [20,21,22], while the 2A protein of Enterovirus 71 cleaves MAVS at multiple amino acid (aa) residues [23]. In addition, hepatitis C virus and porcine reproductive and respiratory syndrome virus infections inhibit MAVS antiviral signaling by NS3/4A serine protease and 3C-like serine protease, respectively, and ultimately facilitate viral replication [24,25]. In addition to MAVS cleavage, some viral infections mediate proteasome degradation of MAVS. Hepatitis B virus protein X was reported to interact with MAVS and promote ubiquitin (Ub)-proteasome degradation [26]. Moreover, Rotavirus protein VP3 and NSP1 protein catalyze the ubiquitination and proteasome degradation of MAVS by different mechanisms [27,28]. Interestingly, Newcastle disease virus, a member of the Paramyxoviridae family, targets MAVS for ubiquitin-mediated degradation through E3 ubiquitin ligase RING-finger protein 5 (RNF5) [29]. These findings promoted us to investigate the molecular mechanism that exists between aMPV/C infection and MAVS expression.

In this study, our data reveal a post-translational mechanism that negatively regulates MAVS during aMPV/C infection, occurring via a ubiquitin-dependent, proteasome-mediated degradation mechanism in Vero cells. Furthermore, our results show that the formation of ubiquitin chains occurs at amino acids 363, 462, and 501 of MAVS and RNF5 targeting MAVS at Lysine 363 and 462 is involved in MAVS degradation in the aMPV/C-infected Vero cells.

## 2. Materials and Methods

### 2.1. Cell and Viral Culture

Vero cells were originally purchased from the American Type Culture Collection (ATCC) and grown in Dulbecco’s modified Eagle’s medium (DMEM; Gibco, NY, USA) supplemented with 2–10% heat-inactivated fetal bovine serum (FBS) (Gibco, Life Technologies, USA), 100 mg/mL streptomycin, and 100 units/mL penicillin at 37 °C and 5% CO_2_.

aMPV/C strain JC was isolated from meat-type chickens with respiratory syndrome as described previously [9]. The virus strain was propagated and titrated by serial dilutions in Vero cells and used at 10^4.5^ of the 50% tissue culture infectious dose (TCID_50_) per 0.1 mL. aMPV/C was inactivated with ultraviolet (UV) at 75 mWs/cm_2_ using a low-pressure mercury vapor discharge lamp and then inoculated onto cultured cells to detect the ability of virus replication [30].

### 2.2. Antibodies and Reagents

The following antibodies were obtained from Sigma-Aldrich: mouse anti-β-actin (A1978), mouse anti-Flag (F1804), horseradish peroxidase (HRP)-conjugated goat antimouse (A9044), and HRP-conjugated goat antirabbit (A0545). Rabbit polyclonal antibodies against MAVS (A5764) and RNF5 (A8351) were purchased from ABclonal Technology. Rabbit polyclonal antibodies against MARCH5 (ab185054) and HA (3724s) were obtained from Abcam and Cell Signal Technology, respectively. Rabbit anti-N polyclonal antibody and mouse anti-N monoclonal antibody were prepared in our laboratory. The proteasome inhibitor MG-132 (S2619, Sellek), the autophagy inhibitor wortmannin (S2758, Sellek), and Chloroquine (CQ) (C6628, Sigma) were used in the experiments. The Enhanced Cell Counting Kit-8 (C0042) and DAPI (C1002) were purchased from Beyotime.

### 2.3. Plasmids Construction

The monkey MAVS gene was amplified from Vero cells with gene-specific primers based on MAVS sequences available in the GenBank database (accession no. NM_001042666, Table 1) and subcloned into p3×FLAG-CMV (Sigma, E7658) to generate the following expression plasmid: pFLAG-MAVS. The following recombinant plasmids of the truncated MAVS gene were constructed: p3×FLAG-CMV-MAVS1, MAVS2, and MAVS3 (aa 1–201, aa 202–359, and aa 360–541, Table 1). FLAG-MAVS3 mutants were generated by site-directed mutagenesis (Rui Biotech Co., Ltd.): FLAG-MAVS3mt1, FLAG-MAVS3mt2, FLAG-MAVS3mt3, FLAG-MAVS3mt4, FLAG-MAVS3mt5 (K363A, K372A, K421A, K462A, and K501A, respectively) and FLAG-MAVS3mt-sim (K363A, K462A, and K501A, simultaneously). All of the above plasmids were confirmed to be correct by sequencing. pRK5-HA-ubiquitin (17608), pRK5-HA-ubiquitin-K48 (17605), and pRK5-HA-ubiquitin-K63 (17606) were purchased from Addgene.

### 2.4. aMPV/C Infection and Virus Titration

Vero cells were incubated with aMPV/C at a multiplicity of infection (MOI) of 0.5 or 0.1, or mock-infected with DMEM. Following a 1.5 h absorption time, unattached viruses were removed and the cells were washed with phosphate-buffered saline (PBS) and then cultured in DMEM supplemented with 2% FBS at 37 °C for the indicated time points in different experiments.

The virus titer was assayed on aMPV/C-infected Vero cells monolayers. Following 1.5 h incubation on Vero cells with serially diluted cell supernatant, fresh medium was added and incubated for five days, cytopathic effects (CPEs) were observed under a microscope and virus titer was calculated as TCID_50_ per 0.1 mL.

### 2.5. Indirect Immunofluorescence Assay

Vero cells with 90% confluence were infected with aMPV/C (MOI = 0.5) in 24-well culture plates. The cells were fixed with precooled and permeabilized using 4% paraformaldehyde (Sigma-Aldrich, 16005), 0.1% Triton X-100 (Sigma-Aldrich, T8787) and in 2% BSA (Beyotime, ST023) in PBS at different indicated time points, and anti-N monoclonal antibody and anti-MAVS rabbit polyclonal antibody were co-incubated with the cells for 2 h at 37 °C. After 3 washes with PBS-Tween-20 (PBST containing 0.05% Tween-20 [Sigma, P1379]), the cells were co-incubated with secondary fluorescein isothiocyanate (FITC)-conjugated antirabbit and Tetramethylrhodamine-6-isothiocyanate (TRITC)-conjugated antimouse antibodies for 2 h at 37 °C. Finally, the cells were washed with PBST and directly observed under an Olympus IX73 immunofluorescence microscope.

### 2.6. RNA Preparation, Reverse Transcription-Polymerase Chain Reaction, and Quantitative Real-Time RT-PCR (qRT-PCR)

Total RNA from aMPV/C-infected or mock-infected Vero cells was isolated with an RNeasy Mini Kit (Qiagen, 74104) according to the manufacturer’s protocol. The cDNA was synthesized with 2 μg of total RNA using the FastKing RT Kit (TIANGEN, KR116-02) as the template, followed by real-time PCR (MAVS and glyceraldehyde-3-phosphate dehydrogenase (GAPDH)) with the gene-specific primers listed in Table 1. Primers for the MAVS gene and GAPDH gene were designed based on sequences available in the GenBank database (accession no. NM_001042666 and NM_001195426, respectively).

### 2.7. Silencing MARCH5 or RNF5 Gene with Small Interfering RNA (siRNA)

Vero cells were transfected with 40 pmol siRNA using Lipofectamine RNAiMAX Transfection Reagent (Invitrogen, 13778150) according to the manufacturer′s protocol. Forty-eight hours after transfection, the cells were lysed for analyzing the silencing efficiency of MARCH5 and RNF5 by Western blotting or were infected aMPV/C for Western blot analysis. The siRNA targeting MARCH5 and RNF5 were designed by GenePharma Company (Suzhou, China): MARCH5-siRNA (sense, 5′-GGGUGGAAUUGCUUUUGUUTT-3′; antisense, 5′-AACAAAAGCAAUUCCACCCTT-3′), RNF5-siRNA (sense, 5′-GUGUCCAGUAUGUAAAGCUTT-3′; antisense, 5′-AGCUUUACAUACUGGACACTT-3′). The Scrambled siRNA (Scra) was as follows: sense, 5′-UUCUCCGAACGUGUCACGUTT-3′; antisense, 5′-ACGUGACACGUUCGGAGAATT-3′.

### 2.8. Cell Transfection, Immunoprecipitation, SDS-PAGE, and Western Blotting

Vero cells were plated on a 6-well culture plate (Thermo Scientific, 3516, Waltham, MA, USA) and transfected with the indicated plasmids for 6 h using Lipofectamine 3000 (Invitrogen, L3000150, Waltham, MA, USA) according to the manufacturer’s protocol, and then infected with aMPV/C for 24, 48, or 72 h.

For immunoprecipitation (IP), cell lysates were prepared in IP buffer (Beyotime, P0013), followed by centrifugation at 12,000× g for 20 min. The supernatants were precipitated with 20 µL anti-Flag Magnetic Agarose (Thermo Scientific, Waltham, MA, USA, A36797) and then gently rocked overnight at 4 °C. The beads were washed five times with IP buffer, followed by boiling with sodium dodecyl sulphate (SDS) loading buffer for 10 min. Proteins eluted from the beads were subjected to SDS-polyacrylamide gel electrophoresis (PAGE) and Western blotting.

Immunoblot analysis was performed as described previously (Hou et al., 2017). Briefly, proteins were separated by SDS-PAGE and transferred to nitrocellulose membranes (PALL, 66485). The membranes were blocked with skimmed milk and incubated with appropriate primary antibodies and horseradish peroxidase (HRP)-conjugated goat antimouse or goat-antirabbit IgG, followed by detection with a SuperSignal West Femto Substrate Trial Kit (Thermo Scientific, 34096) and exposure to an enhanced chemiluminescence apparatus (ProteinSimple, Santa Clara, CA, USA).

### 2.9. MTT Assay

Cell viability was assessed using an Enhanced Cell Counting Kit-8 according to the manufacturer’s instructions (Beyotime C0042). Approximately 4 × 10^4^ Vero cells per well were seeded in a 96-well cell culture plate and cultured overnight at 37 °C under 5% CO_2_. The fresh medium containing MG132, CQ or wortmannin was added, and the cells were incubated for indicated time points. 10 μL of MTT solution were added and incubated for 1–4 h at 37 °C. The optical density was measured at 450 nm using a microplate reader.

### 2.10. Statistical Analysis

The data are presented as mean ± standard deviation (SD). The significance of the variability between different treatment groups was determined by Two-way ANOVA tests of variance using the GraphPad Prism software. *p* < 0.05 was considered statistically significant.

## 3. Results

### 3.1. aMPV/C Infection Induces MAVS Reduction in Vero Cells

To determine whether MAVS is affected in aMPV/C-infected Vero cells, we used Western blotting to examine the change in MAVS expression, which is a critical hallmark of the antiviral response. Expression of MAVS gradually decreased over the 120 h after aMPV/C infection (Figure 1A), whereas the amount of MAVS mildly increased in mock-infected Vero cells (Figure 1B). In addition, the increase in viral N protein was used to track the progression of aMPV/C infection (Figure 1A). As illustrated in Figure 1C, the densitometry ratios of MAVS to β-actin bands decreased after 24 h in the virus-infected cells, it was much lower than that in the mock-infected cells from 48 h post-infection (hpi) onward (*p* < 0.01), indicating that aMPV/C infection induced the reduction in MAVS.

To further confirm that the decrease in MAVS is related to aMPV/C infection, the infected cells at each of the indicated time points was analyzed by IFA. The data from our IFA analysis suggested that, with the extension of virus infection time, the number of cells expressing aMPV/C N protein showed an increasing trend, while expression of MAVS decreased gradually (Figure 1D). Moreover, the growth kinetics of virus in Vero cells showed that the viral titers increased from 24 h to 120 h after aMPV/C infection (1E), and the representative photomicrographs of the CPE at indicated time points are displayed in the Appendix A.

To study whether the decrease in MAVS is associated with the inoculation dose of the virus, we detected changes in MAVS at different inoculation doses. The results showed that the cells infected with aMPV/C at an MOI of 0.5 showed more reduction in MAVS than that at an MOI of 0.1 (Figure 1F), indicating that aMPV/C infection caused the decrease in MAVS protein expression in a dose-dependent manner. These results suggest that MAVS reduction is induced in Vero cells during aMPV/C infection.

UV-inactivated viruses are thought to lose viral infectivity in cultured cells. Thus, to further investigate whether active aMPV/C replication is required for the induction of MAVS reduction, Western blotting was used to detect the change of MAVS in Vero cells infected with UV-inactivated aMPV/C. The complete loss of viral infectivity following UV inactivation was confirmed by assaying the virus titer (data not shown). As shown in Figure 1G, no obvious change occurred in MAVS expression in cells infected with UV-inactivated aMPV/C and in mock-infected cells during virus infection. Meanwhile, no N protein was detected in either condition, indicating that active aMPV/C replication is indispensable for MAVS reduction.

### 3.2. MAVS Degradation Is Recovered in aMPV/C-Infected Vero Cells Treated with MG132

The change of protein expression is mainly attributed to two aspects of regulation: transcriptional levels and post-translational levels. Since it was initially found that the decrease in MAVS expression occurred at 48 h after infection (Figure 1A), we analyzed which regulation of transcription or translation played a major role in the decrease in MAVS. To determine whether the reduction in MAVS occurs at the transcriptional level in aMPV/C-infected Vero cells, the levels of MAVS mRNA were measured by qRT-PCR. As shown in Figure 2A, no significant differences in MAVS levels were observed between aMPV/C-infected cells and mock-infected cells at 48 h after virus infection. Moreover, a similar result was found at different inoculation doses in virus-infected Vero cells (Figure 2A). These results indicate that the mRNA level of MAVS was not affected by aMPV/C infection at 48 h. In other words, the reduction in MAVS induced by aMPV/C infection did not occur at the transcriptional level at 48h after virus infection.

Negative post-translational regulation of MAVS in virus-infected cells involves two processes (proteasomal degradation and autophagy) [31]. MG132, a proteasome inhibitor, was initially added to aMPV/C-infected cells to evaluate whether the proteasome pathway participated in MAVS degradation. The results showed that MAVS degradation by aMPV/C was effectively blocked in the presence of MG132 (Figure 2B,C). Decreased MAVS expression was observed in the aMPV/C-infected cells following CQ (lysosome inhibitor) or wortmannin (autophagy inhibitor) treatment, whereas viral N protein expression was mildly reduced in the presence of CQ or wortmannin (Figure 2D,E). Additionally, the MTT results showed that the viability of cultured cells was not affected by pharmaceutical reagents (MG132, CQ, and wortmannin) (Figure 2F). Taken together, these results demonstrate that addition of MG132 largely inhibited MAVS degradation in the aMPV/C-infected Vero cells, suggesting that MAVS degradation by aMPV/C is related to the proteasome pathway.

### 3.3. aMPV/C Infection Induces MAVS Degradation through Ubiquitination

To determine whether aMPV/C infection can catalyze polyubiquitin chain formation on MAVS, we conducted ubiquitination assays in Vero cells. The cells were co-transfected with Flag-MAVS and hemagglutinin (HA)-Ub plasmids, followed by aMPV/C infection and an immunoprecipitation assay at 48 h. As shown in Figure 3A, aMPV/C infection induced polyubiquitin chains formation and MAVS polyubiquitination. In addition, MAVS polyubiquitination was analyzed at 24, 48, and 72 hpi. The immunoprecipitation results showed that addition of MG132 blocked MAVS degradation and increased MAVS polyubiquitination at different time points in aMPV/C-infected Vero cells (Figure 3B). Collectively, these results indicate that aMPV/C infection promotes the formation of polyubiquitin chains and final degradation of MAVS.

### 3.4. MAVS Is Degraded by K48-Linked Ubiquitination

To determine the pattern of MAVS ubiquitination in aMPV/C-infected Vero cells, we co-transfected Vero cells with plasmids encoding Flag-MAVS or Flag and HA-Ub-K48 or HA-Ub-K63, followed by aMPV/C infection and an immunoprecipitation assay. The data showed that aMPV/C-induced K48-linked ubiquitination of MAVS was stronger than K63-linked ubiquitination (Figure 4). This indicates that MAVS mainly undergoes K48-linked ubiquitination in the presence of MG132 in aMPV/C-infected Vero cells.

### 3.5. Amino Acids 363, 462, and 501 Are Sites of MAVS Ubiquitination

Proteins containing lysine (K) are often ubiquitinated at lysine residues. To identify the lysine residues of MAVS to which ubiquitin was connected, we first determined the critical domain of MAVS responsible for MAVS ubiquitination. Three plasmids of truncated MAVS containing complete lysine sites were constructed. Vero cells were co-transfected with HA-Ub and Flag-truncated MAVS1 (aa 1–201), MAVS2 (aa 202–359), and MAVS3 (aa 360–541), followed by aMPV/C infection and immunoprecipitation. The results showed that MAVS3 can undergo polyubiquitination in the presence of MG132 (Figure 5A).

Sequence analysis revealed only five lysine sites (K363, K372, K421, K462, and K501) in MAVS3. To determine which lysine sites are linked by polyubiquitin chains, we mutated one-by-one with each lysine residue mutated to alanine (K363A, K372A, K421A, K462A, and K501A) and generated five individual mutants of MAVS3 (MAVS3mt1, MAVS3mt2, MAVS3mt3, MAVS3mt4, and MAVS3mt5), which investigated the effect of aMPV/C infection on the ubiquitination of these mutants in Vero cells. As shown in Figure 5B, in the presence of MG132, the MAVS3 mutants K363A, K462A, and K501A partially blocked aMPV/C-induced ployubiquitination, whereas no obvious changes were observed in Vero cells transfected with MAVS3 mutants K372A or K421A. This suggests that residues K363, K462, and K501 in MAVS are sites of aMPV/C-mediated ubiquitination. To confirm these critical ubiquitination sites, we simultaneously mutated the lysine residues 363, 462, and 501 to alanine in MAVS3 and examined the effect on ubiquitination. The data showed that the simultaneous mutant of MAVS3 completely lost its capacity to be ubiquitinated (Figure 5C). Taken together, these results demonstrate that the amino acids 363, 462, and 501 in MAVS are critical sites for forming polyubiquitin chains and are involved in MAVS degradation in aMPV/C-infected Vero cells.

### 3.6. RNF5 Is Involved in MAVS Degradation in aMPV/C-Infected Vero Cells

Our study found that aMPV/C degraded MAVS by forming polyubiquitin chains at lysine 363, 462, and 501. Moreover, a recent report showed that E3 ligase RNF5 or MARCH5 targets MAVS at lysine 363 and 462 or lysine 7 and 501 for K48-linked ubiquitination and degradation [31]. These data prompted us to explore whether RNF5 or MARCH5 play an important role in MAVS degradation. The effect of silencing RNF5 or MARCH5 with siRNA on MAVS expression in aMPV/C-infected Vero cells was assessed. The results showed that Vero cells transfected with 40 pmol siRNA targeting RNF5 or MARCH5 showed an obvious reduction in protein expression (Figure 6A) and the dose of 40 pmol siRNA was used for subsequent experiments. As shown in Figure 6B,C, compared to the siScra-transfected cells, MAVS degradation was restored in aMPV/C-infected Vero cells transfected with RNF5-siRNA and the knockdown of MARCH5 expression had no obvious effect on MAVS degradation, which was different from the mechanism of NDV-induced MAVS degradation [29]. To further explore the effect of RNF5 on K48-linked ubiquitination of MAVS or MAVS3, the effect of silencing RNF5 with siRNA on MAVS or MAVS3 ubiquitination in aMPV/C-infected Vero cells was assessed. The results showed that the K48-linked ubiquitination of MAVS or MAVS3 was significantly reduced in aMPV/C-infected Vero cells with siRNF5 (Figure 6C). Taken together, these results demonstrate that RNF5 targeting MAVS at lysine 363 and 462 is involved in MAVS degradation in aMPV/C-infected Vero cells.

## 4. Discussion

MAVS transmits the upstream activation signal of retinoic acid inducible gene I or melanoma differentiation-associated protein-5 to downstream intracellular pathways signals in type I interferon production, and is part of the antiviral immune response [32]. Increasing research data have shown that MAVS plays a critical role in antiviral immunity. Many reports have focused on the interaction and regulated mechanisms between virus infection and MAVS function [29,33,34]. However, to date, no study has investigated whether MAVS expression is regulated by aMPV/C infection and if so, how it is regulated during viral infection. In this study, we demonstrated that aMPV/C infection induced MAVS degradation via the proteasome pathway in the cultured cells, and this degradation was largely blocked in the presence of MG132. Further analysis revealed that the amino acids 363, 462, and 501 in MAVS were critical sites for forming polyubiquitin chains and RNF5 is involved in MAVS degradation in aMPV/C-infected Vero cells.

Vero cells, an epithelioid cell derived from the kidney of African green monkeys, yield high viral titer and are preferred for the propagation of aMPV [9,35,36]. They are model cells for studying the change in MAVS during aMPV/C infection. Although Vero cells have a genetic defect in interferon production, this defect does not affect the transcription, expression, and function of MAVS in cells [37]. Thus, we focused on the mechanism of aMPV/C-induced change in MAVS in this study.

First, our Western blotting analysis demonstrated that MAVS was significantly decreased in the aMPV/C-infected Vero cells compared to in mock-infected Vero cells (Figure 1A–C). Importantly, from 48 hpi onwards, an obvious reduction in MAVS was observed. Since autophagy and apoptosis are significantly induced in the late stage of viral infection and participate in viral replication [36], it may indirectly affect the degradation of MAVS induced by aMPV/C infection. Therefore, in most experiments, this time point (48 hpi) was mainly chosen to further analyze the mechanism of aMPV/C-mediated MAVS reduction in Vero cells. Notably, a high dose of aMPV/C (MOI = 0.5) induced more MAVS degradation than a low dose (MOI = 0.1) (Figure 1F), indicating that the decrease in MAVS is related to the virus inoculation dose. In addition, the decrease in MAVS was accompanied by an increase in viral N protein expression and no change in MAVS was observed in the UV-aMPV/C-infected cells or mock-infected cells (Figure 1G) indicating that active aMPV/C replication is indispensable for MAVS reduction in Vero cells. In the further study, we used indirect immunofluorescence and virus titer assay to confirm that the reduction in MAVS was related to virus infection in Vero cells. The decrease in MAVS expression was not caused by a low percentage of infected cells or virus-induced massive cell death, but mainly due to the increase in the number of virus-infected cells (Figure 1D). In addition, the increasing trend of virus proliferation curve from 24 to 120 h after aMPV/C infection also indirectly indicates that virus infection induces the decrease in MAVS (Figure 1E). Importantly, we have used defective cells of IFN-I production, Vero cells, for this study, which excludes the possibility that interferon induced the reduction in MAVS expression reported in the literature [38,39].

MAVS expression and function are tightly regulated at both the transcriptional and translational levels. To determine if the reduction in MAVS occurred at the transcriptional or translational level, we quantified MAVS mRNA by qRT-PCR and analyzed protein expression in the cultured cells treated with different pharmaceutical reagents by Western blotting. Our results showed that the level of MAVS mRNA did not change in the aMPV/C-infected cells or mock-infected cells, regardless of the viral inoculation dose (Figure 2A). This indicates that the reduction in MAVS by aMPV/C is not mediated at the transcriptional level. These data are consistent with studies of MAVS reduction during NDV infection [29]. The decrease in protein levels was mainly achieved by protein cleavage and protein degradation. In this study, we observed no cleaved bands of MAVS by Western blotting; therefore, we focused on the mechanism of MAVS degradation in further experiments.

Importantly, the proteasome pathway plays a pivotal role in protein degradation [31]. MG132, a proteasome inhibitor, was added to Vero cells to analyze the change in MAVS expression. Our results demonstrated that MAVS degradation was effectively restored in the aMPV/C-infected Vero cells (Figure 2B,C), suggesting that the proteasome pathway was mainly responsible for MAVS degradation. This is consistent with MAVS regulation by many other viruses [26,27,28,29]. Previous studies showed that autophagy and lysosome are also important pathways for MAVS degradation [40,41] and that aMPV/C infection induced autophagy begins at 48 hpi in Vero cells [36], prompting us to evaluate whether MAVS degradation is related to autophagy and the lysosome in the aMPV/C-infected Vero cells. Our results showed that MAVS was degraded by aMPV/C infection in the presence of wortmannin (autophagy inhibitor) or CQ (lysosome inhibitor) (Figure 2D,E), suggesting that MAVS degradation by aMPV/C was not influenced by autophagy and lysosome pathway. Interestingly, N protein expression was decreased in the aMPV/C-infected Vero cells treated with wortmannin or CQ compared to in the aMPV/C-infected Vero cells, indicating that the inhibition of autophagy decreased viral production. This result is similar to those of studies of aMPV/C-induced autophagy [36]. Importantly, the addition of wortmannin or CQ did not fully restore MAVS content and only slightly decreased the degradation of MAVS, which was not similar to the results after MG132 treatment. These results showed that wortmannin or CQ treatment is not the main factor to degrade MAVS, which further indicated that proteasome pathway played a critical role in MAVS degradation (Figure 2D,E). Taken together, our findings illustrate that MAVS degradation mainly occurred at the translational level rather than at the transcriptional level.

The formation of polyubiquitin chains represents activation of the ubiquitin-proteasome pathway, which has been reported to regulate MAVS expression [42]. The immunoprecipitation results showed that aMPV/C infection promoted the formation of MAVS polyubiquitin chains (Figure 3A,B) and K48-linked ubiquitination (Figure 4) in the aMPV/C-infected Vero cells treated with MG132. Lysine is a critical residue connected to ubiquitin molecules for protein ubiquitination. Through segmented expression and site-directed mutagenesis, we determined that residues 363, 462, and 501 in MAVS are involved in the formation of polyubiquitin chains and MAVS degradation in the aMPV/C-infected Vero cells (Figure 5A–C). This differs from MAVS ubiquitination mediated by infection with NDV, which is another member of the Paramyxoviridae family [29]. This presents a unique event of aMPV/C-induced MAVS degradation.

Interestingly, many E3 ligases have been identified to modulate MAVS expression by K48-linked ubiquitination, such as SMAD ubiquitin regulatory factor 1 or 2 (Smurf1 or Smurf2), atrophin 1-interacting protein 4, MARCH5, and RNF5 [43,44,45,46,47]. In these E3 ligases, MARCH5 and RNF5 catalyze MAVS ubiquitination at lysine 7 and 501 and at lysine 363 and 462 for proteasomal degradation, respectively [31,44,47]. Moreover, aMPV/C induced MAVS ubiquitination at lysine 363, 462, and 501 in our study (Figure 5), which suggested that MARCH5 and RNF5 may be involved in aMPV/C-induced MAVS degradation. Further experiments’ results showed that RNF5, instead of MARCH5, is involved in degradation of MAVS in aMPV/C-infected Vero cells (Figure 6). Although MARCH5 catalyze MAVS ubiquitination at lysine 7 and 501 for proteasomal degradation [31,47] and lysine 501 in MAVS is a critical site in aMPV/C-induced ubiquitination of MAVS in cells, the results showed that MARCH5 has no effect on the degradation of MAVS, indicating that there may be an unknown E3 ubiquitin ligase targeting MAVS at lysine 501 degraded MAVS in aMPV/C-infected Vero cells (Figure 6), which was different from the mechanism of NDV-induced MAVS degradation [29]. In that study, NDV degraded MAVS by RNF5 (lysine 363 and 462), MARCH5 (lysine 7 and 501), and Smurf1 in Hela cells. It is well known that aMPV/C and NDV belong to the members of the family Paramyxoviridae, but their composition and growth characteristics are completely different. For example, V proteins, an accessory protein of NDV, play a pivotal role in NDV-inhibited IFN production [29], while aMPV/C does not have a similar protein. Interestingly, although RNF5 is involved in the degradation of MAVS during aMPV/C or NDV infection, it is important that NDV uses viral V proteins to complete this process [29], while aMPV/C does not, which indicated that aMPV/C had significantly different pathogenesis compared with other members of the family Paramyxoviridae in infected cells. Moreover, this characteristic is destined to have a unique molecular pathogenesis of aMPV in infected cells. This further confirmed a new mechanism of aMPV/C-induced MAVS degradation, and the screening and identification of the unknown E3 ubiquitin ligase or new degradation mechanism need to be further studied. In addition, sequence alignment showed that the amino acid homology of MAVS protein from chicken and monkeys was very low. Thus, analysis of whether chicken MAVS can be degraded by ubiquitination and clarification of the degradation mechanism are also future directions for research, which is helpful to understand the relationship between MAVS and IFN production. Although the effect of IFN on virus replication after MAVS degradation cannot be studied in aMPV/C-infected Vero cells, the study of aMPV/C infection degrading MAVS through ubiquitination enriches the interaction between virus and host cells.

In conclusion, our study demonstrated for the first time that aMPV/C infection mediated MAVS degradation in Vero cells. Our results further showed that polyubiquitin chains formed at lysine 363, 462, and 501 in MAVS and RNF5 participated in aMPV/C-induced MAVS degradation. Therefore, these data provide an important foundation to further study pathogenic mechanism of aMPV/C.

## Figures and Tables

**Figure 1 viruses-13-01990-f001:**
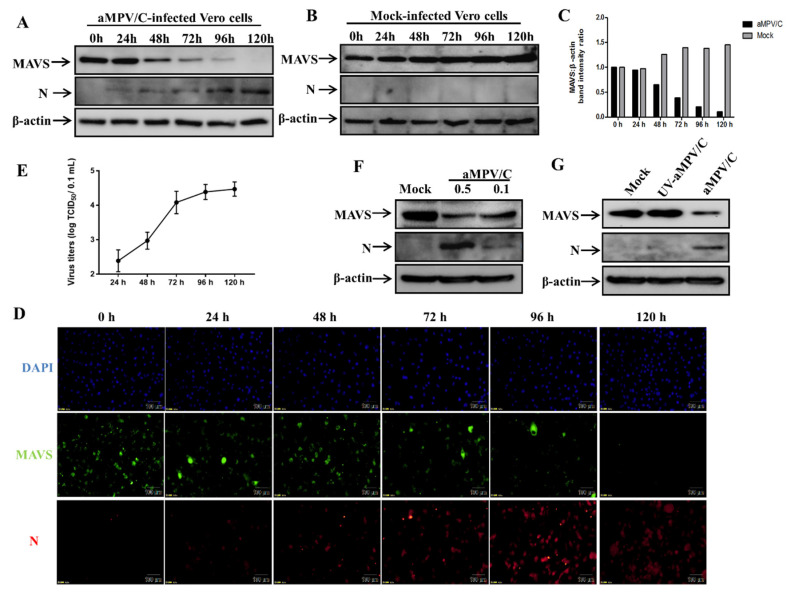
aMPV/C infection induces MAVS reduction in Vero cells. (**A**,**B**) Proteins were extracted from aMPV/C-infected or mock-infected Vero cells at 0, 24, 48, 72, 96, and 120 hpi, and were analyzed by SDS-PAGE and Western blotting with anti-MAVS and antiviral N antibodies. β-actin was used as a protein loading control. (**C**) The relative band densities of MAVS: β-actin normalized to the control conditions. (**D**) aMPV/C-infected or mock-infected Vero cells were subjected to indirect immunofluorescence analysis at 0, 24, 48, 72, 96, and 120 hpi. The MAVS expression signal (green) and viral N protein staining (red) are shown. (**E**) The growth kinetics of aMPV/C in Vero cells was assayed using TCID_50_ (n = 3). (**F**) Vero cells infected with aMPV/C at an MOI of 0.1 or 0.5 were harvested at 48 hpi and analyzed by Western blotting with anti-MAVS, antiviral N, and anti-β-actin antibodies. (**G**) Vero cells were infected with aMPV/C or UV- aMPV/C (MOI = 0.5) for 48 h. Subsequently, proteins were extracted and analyzed as described in A and B.

**Figure 2 viruses-13-01990-f002:**
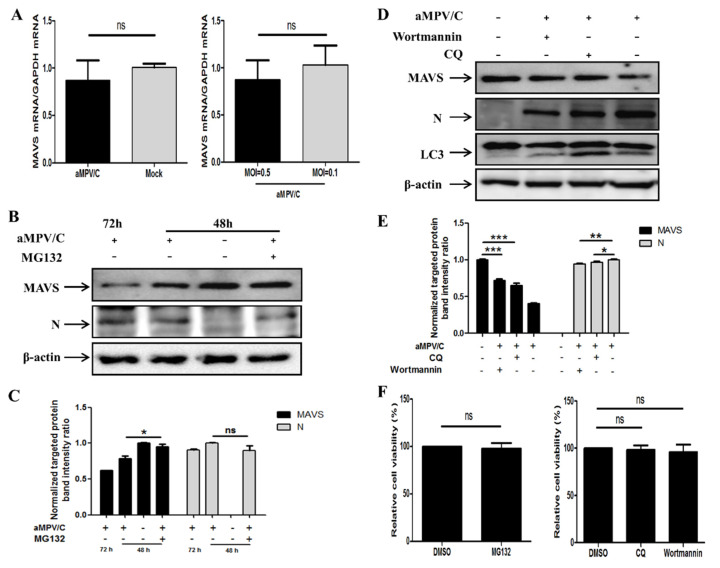
MAVS degradation is inhibited by MG132. (**A**) Total cellular RNA was extracted from aMPV/C-infected (MOI = 0.1 or 0.5) or mock-infected Vero cells at 48 h, and qRT-PCR was employed to analyze MAVS and GAPDH levels. (**B**) aMPV/C-infected (MOI = 0.5) or mock-infected Vero cells were treated with or without MG132 (10 μM) for 12 h prior to cell harvesting at 48 or 72 hpi and protein levels of MAVS and N were detected. The presence of MG132 is indicated with “+.” (**C**) Representative results are displayed with graphs corresponding to the ratios of MAVS: β-actin or P: β-actin normalized to the control conditions. (**D**) aMPV/C-infected (MOI = 0.5) Vero cells were treated with CQ (20 μM) or wortmannin (100 nM), harvested after 48 dpi, and analyzed as described in C. (**E**) Representative results are displayed with graphs corresponding to the ratios of MAVS: β-actin or N: β-actin normalized to the control conditions. (**F**) Then the cell viability upon pharmacological treatments was determined by the Enhanced Cell Counting Kit-8. Representative results are shown in a graph representing the percentage of relative cell viability. Error bars, mean ± SD of three independent experiments. ns (not significant) *p* > 0.05; * *p* < 0.05; ** *p* < 0.01; *** *p* < 0.001, compared with the control group.

**Figure 3 viruses-13-01990-f003:**
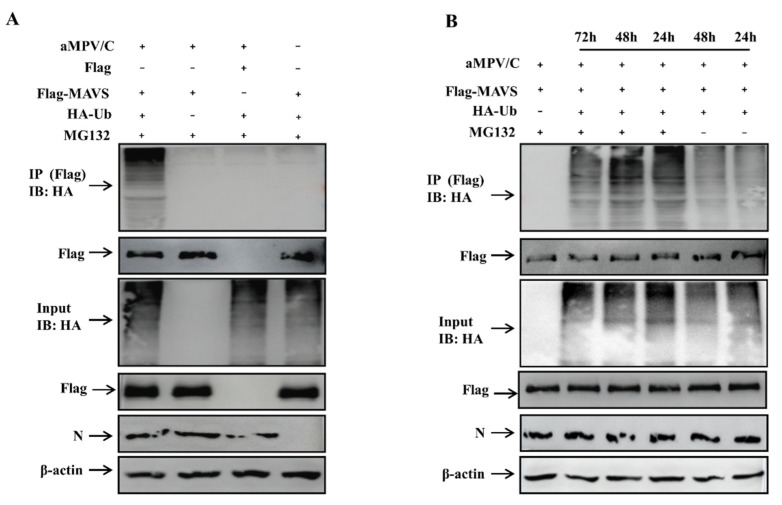
MAVS polyubiquitination is analyzed in aMPV/C-infected or Mock-infected Vero cells. (**A**) Vero cells were co-transfected with Flag-MAVS and HA-Ub. After 6 h transfection, the cells were infected with aMPV/C (MOI = 0.5) for 48 h in the presence of MG132, and analyzed by IP with anti-Flag antibody or Western blot with anti-HA antibody. In addition, Flag, HA and β-actin in aMPV/C-infected cells detected by immunoblotting were used as an input control and an internal loading control, respectively. aMPV/C infection was verified with antiviral N antibody. (**B**) Vero cells were co-transfected with Flag-MAVS and HA-Ub. After 6 h transfection, the cells were infected with aMPV/C (MOI = 0.5) for 24, 48, or 72 h in the presence of MG132, and analyzed as described in A.

**Figure 4 viruses-13-01990-f004:**
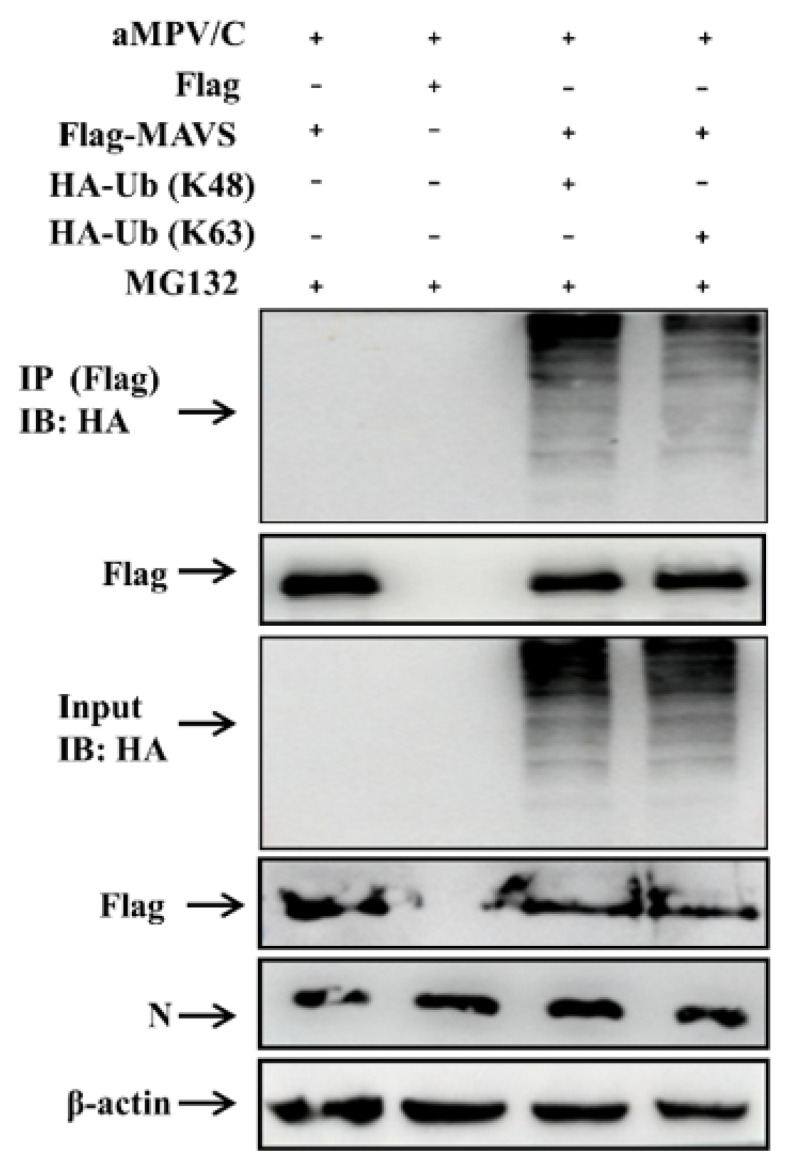
MAVS mainly undergoes K48-linked ubiquitination. Vero cells were co-transfected with Flag-MAVS and HA-Ub (K48) or HA-Ub (K63). After 6 h transfection, the cells were infected with aMPV/C (MOI = 0.5) for 48 h in the presence of MG132, and analyzed using IP with anti-Flag antibody or Western blot with anti-HA antibody. In addition, Flag, HA and β-actin in aMPV/C-infected cells detected by immunoblotting were used as an input control and an internal loading control, respectively. aMPV/C infection was verified with antiviral N antibody.

**Figure 5 viruses-13-01990-f005:**
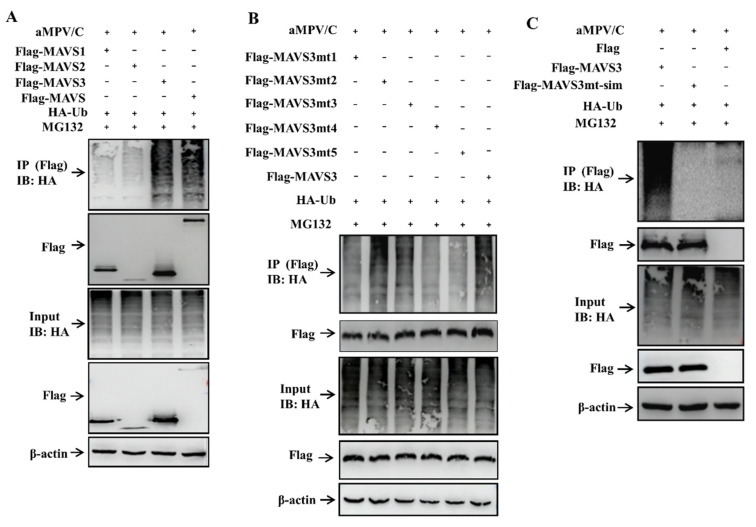
The critical sites for MAVS polyubiquitination are identified. (**A**) Vero cells were co-transfected with Flag-MAVS1, Flag-MAVS2, Flag-MAVS3, or Flag-MAVS and HA-Ub. After 6 h transfection, the cells were infected with aMPV/C (MOI = 0.5) for 48 h in the presence of MG132, and analyzed by IP with anti-Flag antibody or Western blot with anti-HA antibody. In addition, Flag, HA and β-actin in aMPV/C-infected cells detected by immunoblotting were used as an input control and an internal loading control, respectively. aMPV/C infection was verified with antiviral N antibody. (**B**) Vero cells were co-transfected with Flag-MAVS3mt1, Flag-MAVS3mt2, Flag-MAVS3mt3, Flag-MAVS3mt4, Flag-MAVS3mt5 or Flag-MAVS3 and HA-Ub. The cells were infected with aMPV/C and analyzed as described in (**A**). (**C**) Vero cells were co-transfected with Flag, Flag-MAVS3 or Flag-MAVS3mt-sim and HA-Ub. The cells were infected with aMPV/C and analyzed as described in A.

**Figure 6 viruses-13-01990-f006:**
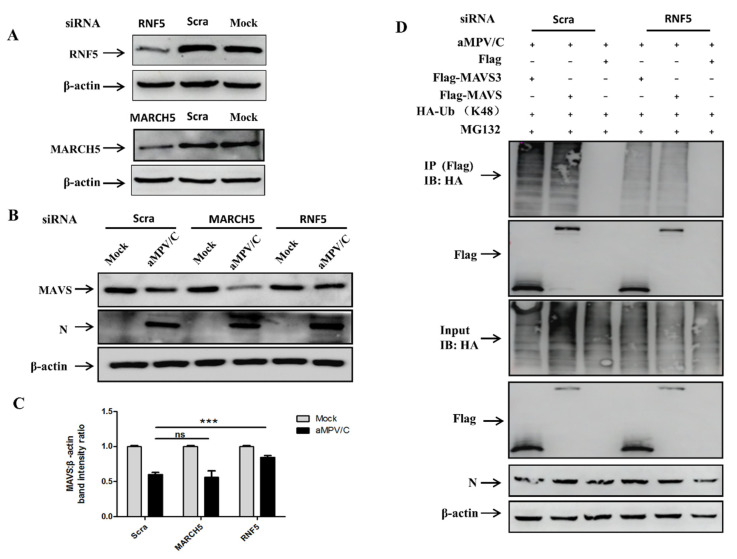
E3 ubiquitin ligases for MAVS degradation are screened and identified in aMPV/C-infected Vero cells. (**A**) Vero cells were transfected with 40 pmol MARCH5-siRNA (MARCH5), RNF5-siRNA (RNF5), and scrambled-siRNA (Scra) and were detected at 48 h by Western blotting with anti-MARCH5, -RNF5, or β-actin antibody. (**B**) Vero cells were transfected with 40 pmol MARCH5-siRNA (MARCH5), RNF5-siRNA (RNF5), or scrambled-siRNA (Scra) and then were infected with aMPV/C for 48 h. Cells were harvested and analyzed using Western blotting with anti-MAVS, anti-N, or anti-β-actin antibody. (**C**) Representative results are displayed with graphs corresponding to the ratios of MAVS: β-actin normalized to the control conditions. (**D**) 40 pmol RNF5-siRNA (RNF5), or scrambled-siRNA (Scra) transfected Vero cells were co-transfected with Flag, Flag-MAVS3, or Flag-MAVS and HA-Ub. The cells were infected with aMPV/C and analyzed as described in A. Error bars represent mean ± SD of three independent experiments. ns (not significant) *p* > 0.05; *** *p* < 0.001, compared with the Scra group.

**Table 1 viruses-13-01990-t001:** Primers and corresponding sequences.

Primers	Sequence (5′–3′)
MAVSF	CAGAATTCGATGCCGTTTGCTGAAGACAAG
MAVSR	TAGGTACCATCTAGTGCAGGCGCCGCCGGTACATCGC
MAVS1F	CAGAATTCGATGCCGTTTGCTGAAGACAAG
MAVS1R	TAGGTACCTATCATTCTGTGTCCTGCTCCTGATG
MAVS2F	ATGAATTCAATGCTGGGCAGTACCCACACAGC
MAVS2R	AAGGTACCTATCACACCATGCCAGCACGGGTTGAGTTGA
MAVS3F	CTGAATTCTATGCCATCCAAAGTGCCTGCTA
MAVS3R	TAGGTACCATCTAGTGCAGGCGCCGCCGGTACATCGC
qMAVSF	CTATAAGTATATCTGCCGCAATT
qMAVSR	AGTCGATCCTGGTCTCTT
qGAPDHF	CAACGGATTTGGTCGTATTGG
qGAPDHR	CGCTCCTGGAAGATGGTG

## Data Availability

Data are contained within the article.

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
