# Peer review of "Avian Metapneumovirus Subgroup C Induces Mitochondrial Antiviral Signaling Protein Degradation through the Ubiquitin-Proteasome Pathway"

_viruses, 2021, doi:10.3390/v13101990_

Round 1
Reviewer 1 Report
Overall
The present work describes the mechanism of MAVS degradation via the ubiquitin-proteasome system during aMPV/C infection, a virus of economic significance. The authors identify two key lysine residues in MAVS that are ubiquitinated during aMPV/C infection. Although these same residues have been previously documented and are likely ubiquitinated during many different infections, this is the first time that aMPV/C has been studied in this way. The authors also find no evidence for an alternative recently described mechanism involving MARCH5, which will also be of interest to the field. Another interesting observation is a possible pro-viral role for autophagy in aMPV/C infection.
I like many aspects of this work. The experiments are generally well designed and the methods well described. The authors’ interpretations are reasonable, even conservative in places, and I enjoyed reading a body of work that was not trying too hard to “sell” itself for a change. The story overall is logical, tight and well referenced, I appreciate the significant amount of effort that the authors have put into this body of work. I have made some suggestions below to try to support the authors, mainly in regards to language and some minor figure alterations, but overall I congratulate the authors on a nice story. I thank you for the opportunity to review this work. Best wishes for your future research.
Major points
- Introduction final paragraph, section headings, and figure caption titles. This could be a matter of personal preference, but there are times in scientific literature when the past tense is appropriate and others when the present feels more appropriate. For example, “We found that / We find that”, “occurred / occurs”, “was involved in / is involved in”. The past tense sometimes hints at a singular or possibly non-reproducible observation rather than the discovery of a biological truth. However, it feels appropriate to use the past tense when describing certain observations or explicit details of experiments (e.g., “Total cellular RNA was extracted…” “qRT-PCR was employed…”, etc). Please consider revising.
- Results, line 194-195 and Figure 1C. The authors state “…the densitometry ratios of MAVS to b-actin bands decreased at 24 h in the virus-infected cells, …”. To my eyes the height of the bars in Figure 1C at 24 h are identical for the aMPV/C-infected and mock-infected, contradicting the quoted statement. Do the authors mean perhaps “after 24 h” or “between 24 h and 48 h” or “at 48 h”? Please check and revise.
- Results, line 194-197 (this is more of a comment than a criticism). The authors compare aMPV/C infected and mock-infected cells and track the expression level of MAVS, concluding that “…aMPV/C infection induced the reduction of MAVS”, another central theme of the manuscript.
One interpretation (I’ll call it interpretation #1) is as stated by the authors above. Another interpretation (interpretation #2) is that the innate immune response is attenuating immune induction, for example, as a normal mechanism of protection against chronic pro-inflammatory signalling and autoimmune damage. Indeed, virus- or interferon-induced interferon expression in as little as 24 h has been reported to reduce MAVS expression (for example, PMID 24569457 Supplemental Excel File 2, PMID 22875473 dataset E-GEOD-37715), although this observation is not shared in other published datasets. Notably, the authors have used Vero cells for this work, which are deficient in IFN-I signalling through a naturally occurring genetic ablation. This would appear to argue against interpretation #2, and I feel it would be beneficial to mention this more strongly in the discussion in support of the authors’ preferred interpretation #1. Another aspect counting against interpretation #2 is the denatured virus used as a negative control, which is great to see, although this virus is presumably incapable of triggering innate immune signalling in Vero and is therefore not the best control in this particular case. The ideal control here would be an aMPV/C strain that is incapable of inducing MAVS degradation. Without such a control, MAVS attenuation as a consequence of innate immune stimulation (i.e., interpretation #2), rather than as a virus-directed phenomenon (interpretation #1), cannot be formally excluded.
- Figure 1, panel D. I can see the overall trends in the images, but these images are quite small and extremely low resolution, even when zoomed-in on a large monitor. Please increase the resolution and/or consider placing the images ‘down’ the page rather than ‘across’ so they can be made much larger. Currently, the images are unfortunately not very useful to the reader.
- Figure 1, panel C. The western blot quantitation in panel C has error bars, indicating multiple samples were analyzed for each time point. If this is the case, please indicate the number of replicates for each time point. If this is not the case, please remove the error bars and state that these data represent n=1.
- Figure 1, panel E. I presume the multiple samples in panel C were quantified in the TCID50 experiment in panel E? If so, please ensure that error bars are indicated on the graph in panel E and indicate the number of replicates (or n=1). Perhaps in this figure or a supplemental figure, please also include representative photomicrographs of the CPE observed over time, if available, so the reader can get an idea of how CPE was scored by the authors for these measurements.
- Line 328. The authors state “Sequence analysis revealed five important lysine sites…”. These are, in fact, the only five lysine residues present in this region of MAVS. The current wording leaves the reader a little mystified as to why these particular five were chosen; was it just blind luck that the authors managed to identify all the acceptor sites? Therefore, I think the authors deserve to be a little bolder with their wording here and should state explicitly that these are the only lysine residues present in this region, that they mutated each of them one-by-one, and this enabled the aMPV/C-induced ubiquitination pattern in this region of MAVS to be fully defined. Nice.
- Figure 5a (another comment). A ubiquitination signal is present with the MAVS3 truncation as well as wild-type, and not with the MAVS1 or MAVS2 truncations. This is a typical approach for most proteins, but MAVS is a little bit unique in that it is only MAVS3 and wild-type that possess the C-terminal transmembrane region required for anchoring into the mitochondrial outer membrane and subsequent immune activation. The absence of a ubiquitination signal in MAVS1 and MAVS2 therefore seems unsurprising (and therefore, possibly a little off-putting to readers who are knowledgeable in this area). Put another way, the apparent requirement for anchoring to the mitochondrial outer membrane would be consistent with MAVS’s activation mechanism and underscore the physiological relevance of the authors’ observations. Therefore, perhaps the authors could find a way to be a little bolder here. Otherwise, how it’s worded currently is alright too.
- Results, all figures. The authors have opted to use the symbol “#” to represent “P > 0.05”. This is fine and the authors are entitled to do this, but I am uncertain as to the advantage. Symbols # and * look rather similar when side-by-side in some of the figures. Would the more standard “* / ns” nomenclature not be a better choice?
Minor points
- Introduction, line 72. One possible interpretation of this sentence is that the authors’ data shows MAVS regulation (in general) occurs post-translationally (and only post-translationally). I am sure this is not what the authors mean to suggest, so I propose revising this sentence to “Our data reveals a post-translational mechanism that negatively regulates MAVS during aMPV/C infection, occurring via a ubiquitin-dependent, proteasome-mediated degradation mechanism in Vero cells”, or similar.
- Methods, line 114. “Virus titeration” should read “virus titration”, please revise.
- Methods, line 153. “…was as follow:” should read “…was as follows:”, please revise.
- Line 195 and 201, and elsewhere. Please maintain consistent nomenclature when describing actin (“B-actin” or “β-actin”).
- Line 202, 205 and 254. There is a random Euro (€) symbol, please revise. One of these symbols (line 205) I think should be the start of the caption for panel E.
- Line 261. “Proteasome” is a protein complex or arguably an organelle, while autophagy is a process. Do the authors mean “Proteasomal degradation and autophagy” to describe these two “processes”?
- Line 368. “Vero cells were transfected with 40 pmol MARCH5-siRNA (MARCH5), RNF5-siRNA (RNF5), and scrambled-siRNA (Scra)…”. I think replacing the word “and” with “or” would be clearer, since each cell line was transfected with only one of these (a or b or c), not all three at once (a and b and c).
- Figure 6C, y-axis. “Intensity band ratio”, please revise to “band intensity ratio” or similar.
Author Response
Point 1: Introduction final paragraph, section headings, and figure caption titles. This could be a matter of personal preference, but there are times in scientific literature when the past tense is appropriate and others when the present feels more appropriate. For example, “We found that / We find that”, “occurred / occurs”, “was involved in / is involved in”. The past tense sometimes hints at a singular or possibly non-reproducible observation rather than the discovery of a biological truth. However, it feels appropriate to use the past tense when describing certain observations or explicit details of experiments (e.g., “Total cellular RNA was extracted…” “qRT-PCR was employed…”, etc). Please consider revising.
Response 1: We would like to thank the reviewer for suggestions on tenses which are greatly helpful for us to improve our manuscript. We have modified them to the appropriate tense in introduction final paragraph, section headings, and figure caption titles in our current version of the manuscript. (Line 75, 76, 199, 210, 249, 272, 299, 317, 325, 341, 352, and 406)
Point 2: Results, line 194-195 and Figure 1C. The authors state “…the densitometry ratios of MAVS to b-actin bands decreased at 24 h in the virus-infected cells, …”. To my eyes the height of the bars in Figure 1C at 24 h are identical for the aMPV/C-infected and mock-infected, contradicting the quoted statement. Do the authors mean perhaps “after 24 h” or “between 24 h and 48 h” or “at 48 h”? Please check and revise.
Response 2: We have changed “at 24 h” to “after 24 h” in our current version of the manuscript. (Line 206)
Point 3: Results, line 194-197 (this is more of a comment than a criticism). The authors compare aMPV/C infected and mock-infected cells and track the expression level of MAVS, concluding that “…aMPV/C infection induced the reduction of MAVS”, another central theme of the manuscript.
One interpretation (I’ll call it interpretation #1) is as stated by the authors above. Another interpretation (interpretation #2) is that the innate immune response is attenuating immune induction, for example, as a normal mechanism of protection against chronic pro-inflammatory signalling and autoimmune damage. Indeed, virus- or interferon-induced interferon expression in as little as 24 h has been reported to reduce MAVS expression (for example, PMID 24569457 Supplemental Excel File 2, PMID 22875473 dataset E-GEOD-37715), although this observation is not shared in other published datasets. Notably, the authors have used Vero cells for this work, which are deficient in IFN-I signalling through a naturally occurring genetic ablation. This would appear to argue against interpretation #2, and I feel it would be beneficial to mention this more strongly in the discussion in support of the authors’ preferred interpretation #1. Another aspect counting against interpretation #2 is the denatured virus used as a negative control, which is great to see, although this virus is presumably incapable of triggering innate immune signalling in Vero and is therefore not the best control in this particular case. The ideal control here would be an aMPV/C strain that is incapable of inducing MAVS degradation. Without such a control, MAVS attenuation as a consequence of innate immune stimulation (i.e., interpretation #2), rather than as a virus-directed phenomenon (interpretation #1), cannot be formally excluded.
Response 3: We would like to thank the reviewer for the positive and constructive comments and suggestions which are greatly helpful for us to improve our manuscript. Interferons (IFNs) are important mediators of immune responses by regulating the expression of many antiviral related genes. It is reported that interferons can reduce the expression of MAVS, but this observation does not exist in our study. Because we use Vero cells, an interferon deficient cells, in our study, we rule out the above possibility. In other words, the decrease of MAVS in aMPV/C-infected cells has no relationship with IFNs in Vero cells. We have added content in the discussion section (Line 458-461) and two references (38, 39) in our current version of the manuscript.
In addition, the detailed mechanism of MAVS decrease in aMPV/C-infected Vero cells are also future directions for research. For example, to analyze which viral protein can act as a bridge to regulate the interaction between RNF5 and MAVS, so as to degrade MAVS by RNF5, which can directly prove that aMPV/C infection induces the reduction of MAVS.
Point 4: Figure 1, panel D. I can see the overall trends in the images, but these images are quite small and extremely low resolution, even when zoomed-in on a large monitor. Please increase the resolution and/or consider placing the images ‘down’ the page rather than ‘across’ so they can be made much larger. Currently, the images are unfortunately not very useful to the reader.
Response 4: In order to make the picture look clearer, we have improved the resolution of images (Figure. 1D). Moreover, we have placed them ‘down’ the page to make the picture larger and easier to observe in our current version of the manuscript.
Point 5: Figure 1, panel C. The western blot quantitation in panel C has error bars, indicating multiple samples were analyzed for each time point. If this is the case, please indicate the number of replicates for each time point. If this is not the case, please remove the error bars and state that these data represent n=1.
Response 5: We have modified the quantitative image in Figure. 1C, which only analyze s and represents the quantitative relationship of bands (MAVS and β-actin) in Figure. 1A and 1B.
Point 6: Figure 1, panel E. I presume the multiple samples in panel C were quantified in the TCID50 experiment in panel E? If so, please ensure that error bars are indicated on the graph in panel E and indicate the number of replicates (or n=1). Perhaps in this figure or a supplemental figure, please also include representative photomicrographs of the CPE observed over time, if available, so the reader can get an idea of how CPE was scored by the authors for these measurements.
Response 6: We would like to thank the reviewer for the positive and constructive suggestions. We have modified the growth kinetics of virus (n=3, at different times) in Figure. 1E, and the representative photomicrographs of the CPE at indicated time points are displayed in supplemental figure (Figure S1). (Line 231-233, 664-671)
Point 7: Line 328. The authors state “Sequence analysis revealed five important lysine sites…”. These are, in fact, the only five lysine residues present in this region of MAVS. The current wording leaves the reader a little mystified as to why these particular five were chosen; was it just blind luck that the authors managed to identify all the acceptor sites? Therefore, I think the authors deserve to be a little bolder with their wording here and should state explicitly that these are the only lysine residues present in this region, that they mutated each of them one-by-one, and this enabled the aMPV/C-induced ubiquitination pattern in this region of MAVS to be fully defined. Nice.
Response 7: We have revised previous inaccurate statements to avoid ambiguity in results section in our current version of the manuscript. (Line 364-367)
Point 8: Figure 5a (another comment). A ubiquitination signal is present with the MAVS3 truncation as well as wild-type, and not with the MAVS1 or MAVS2 truncations. This is a typical approach for most proteins, but MAVS is a little bit unique in that it is only MAVS3 and wild-type that possess the C-terminal transmembrane region required for anchoring into the mitochondrial outer membrane and subsequent immune activation. The absence of a ubiquitination signal in MAVS1 and MAVS2 therefore seems unsurprising (and therefore, possibly a little off-putting to readers who are knowledgeable in this area). Put another way, the apparent requirement for anchoring to the mitochondrial outer membrane would be consistent with MAVS’s activation mechanism and underscore the physiological relevance of the authors’ observations. Therefore, perhaps the authors could find a way to be a little bolder here. Otherwise, how it’s worded currently is alright too.
Response 8: We would like to thank the reviewer for the positive and constructive comments which are greatly helpful for us to improve our manuscript. It is reported that most virus proteins (PMID: 20554965P, PMID: 30460894, and PMID: 31270229) can act as an intermediate protein to interact with ubiquitin protein and MAVS to promote the degradation of MAVS through ubiquitination, suggesting that virus proteins play decisive role. In this study, MAVS3 (aa 360–541) is a crucial region in the ubiquitination degradation of MAVS, which suggests that viral proteins of aMPV/C may interact with MAVS3 and RNF5. This inference is also a direction of our in-depth study.
Point 9: Results, all figures. The authors have opted to use the symbol “#” to represent “P > 0.05”. This is fine and the authors are entitled to do this, but I am uncertain as to the advantage. Symbols # and * look rather similar when side-by-side in some of the figures. Would the more standard “* / ns” nomenclature not be a better choice?
Response 9: The symbol “#” has been replaced by “ns (not significant)” to represent “P > 0.05” in results, figures and figure legends in our current version of the manuscript. (Line 284, 419)
Point 10: Introduction, line 72. One possible interpretation of this sentence is that the authors’ data shows MAVS regulation (in general) occurs post-translationally (and only post-translationally). I am sure this is not what the authors mean to suggest, so I propose revising this sentence to “Our data reveals a post-translational mechanism that negatively regulates MAVS during aMPV/C infection, occurring via a ubiquitin-dependent, proteasome-mediated degradation mechanism in Vero cells”, or similar.
Response 10: We have revised corresponding expression content in in our current version of the manuscript. (Line 72-74)
Point 11: Methods, line 114. “Virus titeration” should read “virus titration”, please revise.
Response 11: “Virus titeration” has been changed to “Virus Titration” in the current version of the manuscript. (Line 123)
Point 12: Methods, line 153. “…was as follow:” should read “…was as follows:”, please revise.
Response 12: “was as follow” has been changed to “was as follows” in the current version of the manuscript. (Line 163)
Point 13: Line 195 and 201, and elsewhere. Please maintain consistent nomenclature when describing actin (“B-actin” or “β-actin”).
Response 13: “B-actin” has been changed to “β-actin” in the current version of the manuscript. (Line 217)
Point 14: Line 202, 205 and 254. There is a random Euro (€) symbol, please revise. One of these symbols (line 205) I think should be the start of the caption for panel E.
Response 14: The symbol “€” has been deleted or changed to “E” in the current version of the manuscript. (Line 218, 221, and 280)
Point 15: Line 261. “Proteasome” is a protein complex or arguably an organelle, while autophagy is a process. Do the authors mean “Proteasomal degradation and autophagy” to describe these two “processes”?
Response 15: The meaning expressed in the results section refers to two processes, so “proteasome” has been changed to “proteasomal degradation” in the current version of the manuscript. (Line 287)
Point 16: Line 368. “Vero cells were transfected with 40 pmol MARCH5-siRNA (MARCH5), RNF5-siRNA (RNF5), and scrambled-siRNA (Scra)…”. I think replacing the word “and” with “or” would be clearer, since each cell line was transfected with only one of these (a or b or c), not all three at once (a and b and c).
Response 16: “and” has been changed to “or” in the current version of the manuscript. (Line 410)
Point 17: Figure 6C, y-axis. “Intensity band ratio”, please revise to “band intensity ratio” or similar.
Response 17: “Intensity band ratio” has been changed to “band intensity ratio” in Figure.2A, 2C, 2F, and 6C in the current version of the manuscript.

Reviewer 2 Report
In this article, Hou and colleagues show that avian metapneumovirus subgroup C (aMPV/C) antagonizes the MAVS signaling pathway via post-translational reduction in MAVS protein levels. Specifically, they show that RNF5 targets ubiquitinates MAVS during aMPV/c infection, resulting in MAVS proteasomal-mediated degradation. Overall, the manuscript is very well-written, and the data support their conclusions. Therefore, I recommend this article for publication in Viruses.
Minor comments:
Microcopy images in Fig. 1D are hard to see. Perhaps an inset to zoom in would be helpful. Also, brightening the images would help.
In Figure 2C, there are two bars for MPV/C+ MG132-. Perhaps, these are different time points, but this should be noted on the graph.
Author Response
Point 1: Microcopy images in Fig. 1D are hard to see. Perhaps an inset to zoom in would be helpful. Also, brightening the images would help.
Response 1: In order to make the picture look clearer, we have improved the resolution of images (Figure. 1D). Moreover, we have placed them ‘down’ the page to make the picture larger and easier to observe in our current version of the manuscript.
Point 2: In Figure 2C, there are two bars for MPV/C+ MG132-. Perhaps, these are different time points, but this should be noted on the graph.
Response 2: We have modified the graph in Figure. 2C, and added annotation of corresponding time points in different treatment groups.
